# Insomnia and Information and Communication Technologies (ICT) in Elderly People: A Systematic Review

**DOI:** 10.3390/medsci7060070

**Published:** 2019-06-15

**Authors:** Arianna Salvemini, Grazia D’Onofrio, Filomena Ciccone, Antonella Greco, Anita Tullio, Filomena Addante, Daniele Sancarlo, Gianluigi Vendemiale, Gaetano Serviddio, Francesco Ricciardi, Francesco Giuliani, Antonio Greco

**Affiliations:** 1Geriatric Unit & Laboratory of Gerontology and Geriatrics, Department of Medical Sciences, IRCCS “Casa Sollievo della Sofferenza”, San Giovanni Rotondo, 71013 Foggia, Italy; arysalvemini@virgilio.it (A.S.); filomenaciccone@yahoo.it (F.C.); antonellagreco1986@gmail.com (A.G.); anita.tullio@live.it (A.T.); filenadda74@yahoo.it (F.A.); d.sancarlo@operapadrepio.it (D.S.); a.greco@operapadrepio.it (A.G.); 2The BioRobotics Institute, Scuola Superiore Sant’Anna, 56025 Pontedera, Italy; 3Department of Medical and Surgical Sciences, University of Foggia (C.U.R.E. University Centre for Liver Diseases Research and Treatment), 71122 Foggia, Italy; gianluigi.vendemiale@unifg.it (G.V.); gaetano.serviddio@unifg.it (G.S.); 4ICT, Innovation & Research Unit, IRCCS “Casa Sollievo della Sofferenza”, San Giovanni Rotondo, 71013 Foggia, Italy; f.ricciardi@operapadrepio.it (F.R.); f.giuliani@operapadrepio.it (F.G.)

**Keywords:** insomnia, cognitive behavioural therapy insomnia (CBTi), information and communication technologies, quality of life

## Abstract

**Background:** Insomnia seems to be related to disability, risk of injury, metabolic syndrome, risk for cardiovascular diseases, cognitive impairment, depression and impaired quality of life. **Objectives:** The goals in this paper was (1) to keep track of technological concepts and approaches to improve insomnia in elderly people, and (2) to define the effect that information and communication technologies (ICT) is having on patients’ care. **Design:** A systematic review was conducted from existing literature. Our selection criteria included: (1) age ≥ 60 years; (2) diagnosis of insomnia with the International Classification of Sleep Disorders (ICSD-II), (3) CBTi (cognitive behavioural therapy insomnia), (4) use of technological tools, and (5) associations between insomnia-related variables and indices of disability, quality of life, and global clinical assessments. **Data analysis:** 11 articles were included. An inductive content analysis was used for data extraction. **Results:** Our review revealed any technological systems that could purportedly rehabilitate elderly patients with insomnia. Three categories of research were identified from the review: (1) Internet Deliver-CBTi, (2) virtual coaches, and (3) sleep technologies. **Conclusions:** The potential for ICT to support insomnia care at home can improve the quality of life for families and reduce health care costs and premature institutional care.

## 1. Introduction

Older people suffer from sleep disorders, with over 50% have difficulty falling asleep and sleeping [1]. There are several factors of sleep disturbance in the elderly, such as changes in circadian rhythms, advancing age with chronic medical conditions, and psychosocial changes usually related to aging [2,3,4,5,6]. In addition to being the most at risk for sleep disorders, the elderly are likely to have a clinical diagnosis of insomnia [7]. The World Health Organization (WHO) defines insomnia as a condition of the individual that has one of the following problems: difficulty falling asleep, difficulty in maintaining sleep, early awakening and disturbed sleep for ≥ 2 weeks [8].

Among elderly people, a growing number of studies demonstrate associations between insomnia-related variables and indices of disability. Insomnia in elderly people is of particular concern because it could increase the risk of injury [9], impairs quality of life [10], and may lead to cognitive impairment [11], depression [12], and metabolic syndrome [13]. Moreover, insomnia is associated with a moderately increased risk for cardiovascular diseases [14,15].

However, if insomnia contributes to clinical and functional decline, prevention or treatment of insomnia may prevent or reduce late-life disability [16].

Most treatment guidelines recommend that non-pharmacologic approaches to insomnia control, including sleep hygiene and behavioural methods, can be used as supportive therapies [17,18,19,20]. Among these the cognitive behavioural therapy for insomnia (CBTi) has strong empirical evidence [21]. CBTi demonstrates comparable efficacy with more durable long-term maintenance of gains after treatment discontinuation in randomized controlled trials of direct comparisons of CBTi with sleep medication [19,22,23]. It has been proposed that the skills learned in CBTi can be implemented by the patients on their own beyond discontinuation of CBTi treatment, whereas medication use needs to continue in order to retain the benefit.

Cognitive behavioural therapies refer to a group of techniques that analyse factors that help to perpetuate chronic insomnia, regardless of the cause. Stimulus control therapy provides that insomnia is a maladaptive response to factors such as bedtime and room environment (e.g., regular reading or watching television in bed instead of sleeping [24]. Sleep restriction therapy people with insomnia increase their sleep time by inducing a temporary sleep deprivation with a voluntary reduction of bed time [25]. Relaxation therapies associate insomnia with hyper-excitement [26] therapies requires patient education on sleep needs, correction of expectations and analysis on anxiety and catastrophic thinking, for example an exaggeration on the consequences of reduced sleep. The noise in the bedroom and the use of caffeine are extrinsic factors that can increase insomnia and these factors combine an education in sleep hygiene [27]. The randomized trials have compared cognitive behavioural therapy with drug therapy and combined therapy (cognitive behavioural therapy plus drug therapy). Shorter sleep latency with triazolam at two weeks, but equal four-week latency times, was demonstrated by a study comparing the efficacy of triazolam with cognitive-behavioural therapy [23]. The efficacy of zolpidem with cognitive behavioural therapy has been demonstrated by another study [28]. Significant benefits have been shown for cognitive behavioural therapy groups with a four- to six-week follow-up. A meta-analysis that compared cognitive behavioural therapy studies with those of hypnotics showed similar short-term outcomes during the treatment, with the difference that cognitive behavioural therapy has resulted in a greater reduction of sleep latency [29]. A combination of cognitive behavioural therapy and drug therapy with cognitive-behavioural therapy alone has been compared by several studies [28,30,31]. From the analysis of the studies to 10–24 months of follow-up, it was shown that the improvements were made only thanks to the cognitive-behavioural therapy but not for the combined therapy. This explains why patients are less engaged in learning and practicing cognitive behavioural therapy techniques if they can control insomnia with medications. In contrast, cognitive behavioural therapy, which was established to try to reduce doses of benzodiazepines for patients with chronic insomnia, gave a higher percentage of drug-free patients [32,33]. In 1999, the American Academy of Sleep Medicine published practical techniques for non-pharmacological treatment of chronic insomnia [34]. Stimulus control therapy, progressive muscle relaxation, biofeedback, sleep restriction therapy and multicomponent cognitive behavioural therapy have been recommended. Insufficient evidence is available to recommend education in sleep hygiene, figurative training or cognitive therapy as single therapies. The report of a June 2005 National Institutes of Health sponsored insomnia conference shows that both cognitive behavioural therapy and benzodiazepine receptor agonists are effective in treating insomnia, but that the long-term efficacy of agonists requires further study [35].

There have been significant innovations in recent years in the application of information and communication technologies (ICT) in support of health care for elderly patients. ICT can offer a great deal of potential and can make a very significant difference to the lives of the elderly and their main caregivers. However, the overall opportunities that technology could create for the elderly with insomnia are not clear to date.

The objectives in this paper is to review (1) the application of technological concepts and approaches to improve insomnia in elderly people, and (2) to define the effect that the ICT is having on patients’ care and how healthcare services are organized for elderly people with insomnia.

## 2. Material and Methods

This systematic review was performed in line with the quality requirements of the PRISMA-P guideline [36].

Objectives of the study, previous systematic reviews using qualitative data and recommendations on best practices in the research literature were sources of analysis and research strategies [37,38].

Literature searches were conducted of the MEDLINE, PubMed, Scopus, EMBASE, CINAHL, Web of Knowledge and ScienceDirect databases until November 2016. The search queries included the following terms: [Information and Communication Technologies or ICT], combined with terms to determine the results of interest: [sleep and (disorder OR disorders) App OR Application], and were limited to human studies.

The articles are in English because of the few resources for translation. The reference lists of included articles and relevant review articles were examined to identify any studies lost in electronic research.

The abstracts retrieved from the electronic search were reviewed by a single reviewer to identify the articles that deserve a full review. Before the data were extracted from the relevant documents the complete articles were examined.

The inclusion/exclusion criteria used for our review protocol are as follows.

Inclusion criteria: (1) age ≥ 60 years; (2) diagnosis of insomnia with International Classification of Sleep Disorders (ICSD-II), (3) CBTi (cognitive behavioural therapy insomnia) including sleep restriction, stimulus control, cognitive restructuring, sleep hygiene education and relapse prevention, (4) use of technological tools to improve insomnia, and (5) associations between insomnia-related variables and indices of disability, quality of life, and global clinical assessments.

Exclusion criteria: (1) no English editing (as we lacked resources for translation), and (2) diagnosis of non-insomnia.

No restrictions were made on the grounds of disease duration or drug treatment.

A total of 362 articles, reports and reviews have been identified. After an examination of the abstracts, 166 abstracts were excluded according to the aforementioned inclusion/exclusion and duplication criteria (totalling 179). Six others were excluded after a more in-depth examination (on the basis of the same inclusion/exclusion criteria). Thus, 11 published studies were suitable for the current review (Figure 1). The classification of the articles was done by three authors using an inductive approach to the analysis following different steps for data extraction. First, a preliminary categorization was made, and then the categories were divided among the authors according to their skills. At least one author and the lead author have thoroughly examined each category:Internet delivered – CBTi;Virtual coaches; andSleep technologies.

In reviewing each sheet, the categories have been refined. A detailed summary of each study was provided by the co-authors, including its strengths and weaknesses, as well as an overall rating of the category [39,40]. Through this process: approach, methodology, transparency and strengths and weaknesses were identified. Studies were organized according to theme, in order to provide an overview of the state of the field as a whole. As part of the analysis, the qualitative evaluation of the data was carried out which was intrinsic to the objectives of the review itself.

## 3. Results

Table 1 shows the potential ICTs that support older people with insomnia, with a summary of the 10 published studies selected for this systematic review.

### 3.1. Internet-Delivered CBTi

Recently, development of the web has allowed effective behavioural medicine treatments. The Internet is a useful instrument to provide more appropriate treatment for the patient, cost savings and better availability. At the same time, we have detected some limits linked to CBTi supplied by the Internet, such as individualization of treatment for difficult patients, weakness of provider assistance and leading [41].

Ritterband and others have begun to explore this area [42]. In a randomized pilot study, 45 people were randomly drawn to a wait list control group or to obtain a CBTi Internet intervention: Sleep Healthy Using Internet (SHUTi). The CBTi interventions comprise sleep deprivation, impulse control, cognitive restructuring, sleep hygiene training, fallout prevention and disclosing a significant reduction in the Sleepiness Index in the CBTi. The Internet employment has not produced significant changes to waiting lists. However, there was a reduction of awakening after sleep onset and an improvement of the self-reported quality of sleep, in a six-month follow up. The study notes that an Internet-based approach involves the participants more [43].

SHUTi has tested on people with psychological or medical comorbidities [44].

There is significant evidence, founded on over 20 years of randomized controlled trials (RCTs), for a particular type of cognitive behavioural therapy, CBTi, being an effective and long-term treatment for 50–70% of patients [45,46]. CBTi takes out factors that sustain insomnia over time, including habits perpetuating insomnia, sleep related dysfunctional cognitions and homeostatic sleep drive impairment. Very significant is recording and reading sleeping diaries. The efficiency of CBTi improving outcomes, in patients with primary and comorbid insomnia, has been very often proved, underlining improvements in physical and mental health too outcomes [47,48,49]. Online therapy is also indispensable to deal to the competition from the great range of Internet resources for health and using the most recent Web technology. The target of therapies is in monitoring the results by evaluating evidence-based parameters, such as total sleep time (TST) and sleep efficiency (SE).

To be valid, every online therapy must be flexible and must replicate the interaction with a therapist in person as much as possible. A new interactive video-based online CBTi program was developed by Kirstie N. Anderson et al [50]. This program is founded on screening for other sleep disorders and mental health problems before starting treatment. In this way they demonstrated that the primary target of SE, the sleep latency (SL) and TST were greatly improved. The initial screening permits ruling out people with other sleep disorders (such as restless legs syndrome) or stress, anxiety and depression. Once participants presenting insomnia started therapy, the compliance with therapy instructions and sleep diaries was high, and the attrition rate was low. An improvement of sleep quality using the Pittsburg Sleep Quality Index (PSQI) score and according with the opinion of patients was found. A difference in the way the participants used this system, in the support required and in the sleep of improvement derived, was also found. Using the Internet to obtain an online version of CBTI improves SE and sleep quality more generally [43,51,52,53,54,55]. Consequently, an Internet-based CBTi program showed to be a useful and cheap first step in insomnia treatment. A recent systematic review on this issue evaluated six published RCTs [53], and found there was one further UK RCT published [52].

### 3.2. Virtual Coaches

Although CBTi is effective, there is a lack of knowledge and accessibility regarding this type of therapy. General practitioners are often not aware of the existence of CBTi, and neither is the general public [56]. In addition, there are too few sleep therapists to help all people with insomnia [57]. In order to increase the availability and accessibility of CBTi, Espie et al. suggested a stepped model with Internet-based treatment as a first option [52]. A meta-analysis of computerized CBTi (CCBT-I) concluded that this therapy is a moderately effective self-help intervention for insomnia [53]. Nonetheless, adherence to insomnia and other technology-mediated treatments is often mentioned as a serious problem [58,59,60]. The WHO recognizes the importance of adherence to health regimes in general. They stated, “Adherence is a primary determinant of the effectiveness of treatment” [61]. Various authors, for example, Beun and Donkin, mention that treatment adherence is a problem for cognitive behavioural therapy in general [59,60]. Reports about adherence to various Internet-based interventions show mixed results. For example, Eysenbach gives a few examples in his “law of attrition” of Internet-based interventions with adherence rates ranging from 1–35% [61]. Interestingly, a meta-analysis of CCBT-I reported an average adherence rate of 78% for the six studies they included [53]. However, they did not make a distinction between treatment adherence and experimental compliance, that is, the proportion of the experimental assessments, such as questionnaires, that are completed. Thus, decisive conclusions on the exact adherence rates cannot be made. 

The Sleepcare project aims at the development of a virtual sleep coach that delivers personalized, automated sleep therapy via a mobile phone [62,63]. A key challenge of this e-coach is to provide therapy support in such a way that the coachees really adhere to the regimen of the personal therapy plan. The first step in the development of a virtual sleep coach that meets this adherence challenge is the analysis of current adherence rates, current adherence-enhancing strategies, and coachees’ willingness to accept those strategies. This complementary analysis approach provided new insights on how a virtual coach can support coachees to adhere to sleep therapy (i.e., the needs and constraints). In the study of Corine et al., the envisioned coach would use different adherence-enhancing strategies during the entire coaching process. For example, different roles (e.g., motivator and educator) could be played by different virtual characters to increase the effect of the to-be-developed sleep coach (i.e., split-persona effect) [64]. Around 25 strategies were allocated to the coach ranging from strategies involving others (e.g., peers or family members), helping with planning (e.g., setting goals and making commitments) and gaming strategies (e.g., earning points and taking a quiz). These adherence-enhancing strategies were scripted explicitly in the scenarios in order to discuss them in the focus groups. Various design principles for a virtual sleep coach can be adopted from the interviews and focus groups. The first design principle covers functionality. During the first usage phase, the sleep coach should immediately tickle users’ interest and engage them, for example, by providing automatic sleep tracking. In the interviews, it appeared that interest made coaches start using products. Next, the sleep coach can provide an already-needed functionality (e.g., an alarm clock). According to the interviews, a needed functionality ensures that users keep using a product. Lastly, reminders need to be a part of the sleep coach. Reminders make sure that users do not simply forget to adhere to the coach. Both the participants in the interviews and focus groups indicated that sometimes they just forgot to use a product. Participants in the focus groups showed a positive attitude toward reminders as long as the users were in control over the reminders. Therefore, including reminders in a sleep coach would be a good first step in future research to increase adherence.

A second design principle could be to withhold adherence support at the start of the intervention (i.e., to postpone possible help by a virtual sleep coach). In this way, the coachees are acknowledged and respected as serious, motivated, and autonomous people. Coachees can prove that they adhere to the assignments of the sleep coach; however, the virtual coach can detect when coachees fail to do their assignments, and then offer support. This support can take different forms (reminders, compliments, awarding points, etc.) and can be varied over time based on the needs of the coachee.

A third design principle that can be applied is explaining why willpower does not guarantee success. After such an explanation, the understanding of the added value and acceptance of adherence-enhancing strategies might increase. On top of that, users could be given the control over the employment of adherence-enhancing strategies.

In the authors’ opinion, the most important overall design principle is balance. Coachees should not feel overwhelmed with adherence-enhancing strategies, but appreciate some occasional support. Customization of the virtual sleep coach can ensure that the perfect balance is reached for each and every user. For example, some users might need and appreciate reminders for filling out a sleep diary every day, while other users are more likely to forget to do their relaxation exercises. In this study the treatment adherence seems to be important for the effectiveness of technology-mediated insomnia treatments. Individuals expect that they will adhere well to such treatments and would not gain much from adherence-enhancing strategies. They believe willpower is an effective adherence strategy. The 52% average treatment adherence reported in this paper, however, suggests that there is room for improvement. A virtual coach should be able to cope with this “adherence bias” and persuade users to accept adherence-enhancing strategies (e.g., reminders, compliments and community building). Future research is needed which might help to realize a substantial improvement [65].

### 3.3. Sleep Technologies

According to a recent review (Ko et al.), sleep-related consumer technologies can be differentiated on delivery platform like applications downloaded on mobile device and integrated with the operating system or linked with camera or microphone, wearable devices on the body or on the clothes, embedded in the device or in the sleep environment, computer resources or websites. The aim of these devices is, first of all, to simplify the start of sleep and awakening, self-guided sleep evaluation, sleep education, entertainment, social links and information sharing.

NOWAPI^®^ [66] is a new device designed to remotely monitoring CPAP for get better the quality of sleep, awake fullness, and reduces related risks. It is connected to a CPAP circuit and precisely detects flow variations and breathing cycles and sends data every day or on demand by a GPS/GPRS system with a simple computer connection. Patients enrolled in this study showed full satisfaction and the device was well tolerated [66].

Thus, the new sleep technologies change day-by-day sleep health and sleep medicine is able to improve or injure collective and individual sleep health according to the kind of implementation [67].

## 4. Conclusions

Technological applications are able to rehabilitate the elderly with insomnia with a reduction of the caregiver burden, an improvement of the quality of life for families and the reduction of health care costs. Finally, the potential for technologies to support home insomnia assistance can reduce health care costs by lessening the needs of formal care and premature institutional care. However, the possibility that technology meets the need for insomnia assistance depends on a number of factors, including raising awareness about available technologies and their usefulness, promoting accessibility and convenience and overcoming the challenges to acceptance and use. The possible limitations of the study lie in the fact that there are studies that have not been presented in the review, particularly if reported in the grey literature. However, the sample derived from our review process represents the field in its current state. In future research, in order to improve the use of advanced technologies to be integrated with the current care of insomnia, rigorous clinical trials and continuous technological developments are required.

Moreover, a research agenda for future studies can be developed, as shown below:(1)Focusing on demonstrating the effect of technological approach on sleep quality is important for future studies in this field.(2)Research efforts are needed to develop new technological tools with greater benefits to reduce insomnia and abnormal stress responses.(3)There is a need for assistance intended for caregivers and staff of residential facilities to promote better quality sleep at night (i.e., restrict time spent in bed, minimize daytime napping, provide opportunities for exposure to daylight and exercise).

## Figures and Tables

**Figure 1 medsci-07-00070-f001:**
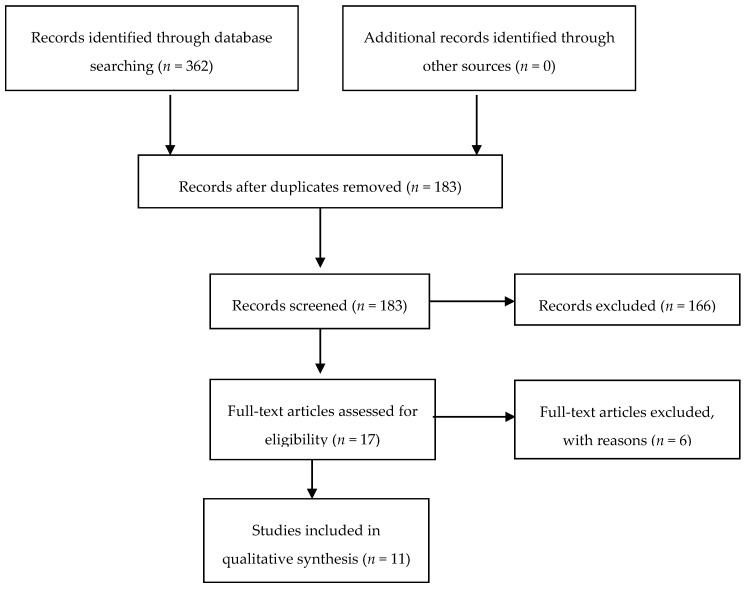
Flow diagram outlining the selection procedure to identify articles which were included in the systematic review of insomnia and assistive technologies in elderly people.

**Table 1 medsci-07-00070-t001:** Current use of information and communication technologies (ICT) for insomnia in elderly people.

Studies	Methods	Outcomes
Ström L et al., 2004	IBTi	Improve total sleep time, total wake time in bed and sleep efficiency
Morin CM V et al., 2005	SHTi	Significant but modest improvements were obtained on subjective sleep parameters
Thorndike FP et al., 2008	SHUTi	Improve sleep, sleep efficiency and overall quality of life
Ritterband LM et al., 2009	IBBI	Considerable potential in delivering a structured behavioural program for insomnia
Kapella MC et al., 2011	CBTi	Significant positive effects were noted in people with COPD for insomnia severity, global sleep quality, wake after sleep onset, sleep efficiency, fatigue, beliefs, attitudes about sleep and depressed mood
Pigeon WR et al., 2012	CBTp + CBTi	Improve sleep, disability from pain, depression and fatigue
Lancee J et al., 2012	EPPC	EPPC were superior compared to the waiting-list condition on most daily sleep measures, global insomnia symptoms, depression and anxiety symptoms
Espie CA et al., 2012	Web-based CBT	Improve the sleep and associated daytime functioning of adults with insomnia disorder
Beun RJ et al., 2014	Sleepcare project	Improve the individual’s adherence to exercises for insomnia therapy
Ko PT et al., 2015	CST	Facilitation of sleep induction or wakening, self-guided sleep assessment, entertainment, social connection, information sharing and sleep education
Leger et al., 2016	NOWAPI^®^	Monitor duration of CPAP which improves the quality of sleep.

**IBTi**: internet-based treatment for insomnia; **SHTi**: Self-help treatment for insomnia; **SHUTi**: Sleep Healthy Using The Internet; **IBBI**: Internet-based behavioural intervention; **CBTi**: CBT for insomnia; **COPD**: chronic obstructive pulmonary disease; **CBTp**: CBT for pain; **EPPC**: electronic and paper-and-pencil conditions; **CBT**: cognitive behavioural therapy; **CST**: Consumer sleep technologies; **CPAP**: Continuous Positive Airway Pressure.

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
