# Peer review of "Insomnia and Information and Communication Technologies (ICT) in Elderly People: A Systematic Review"

_medsci, 2019, doi:10.3390/medsci7060070_

Round 1
Reviewer 1 Report
Please see comments on the paper.

Author Response
1. Pag. 2 – Line 73: May be used in combination
1. Yes. The cognitive behavioral therapy and drug therapy were used in combination and compared with the single therapies. According to the reviewer comment, we have specified better this concept.
2. Pag. 2 – Line 95: I do not feel that this summary belongs in the introduction.
2. The summary in question is a clinical snapshot of the treatment and prevention researches for insomnia, and opens the topic about the technologies used for improving and reducing of the insomnia.
3. Pag. 3 – Line 105: On-line
3. We have not understood the reviewer comment. Maybe, is it referred to a reference to add about PRISMA-P guideline? If yes, we added the required reference.
4. Pag. 6 – Lines 226-228: I don't understand what the authors are trying to convey by this sentence.
4. According to the reviewer comment, we have decided to remove the sentence because it was not clear.
5. Pag. 6 – Line 241: not a word I would use in an academic journal – engage.
5. We added the words “engage them” in the sentence, according to the reviewer observation.
6. Pag. 8 – Lines 293-295: Poor English - needs revising.
6. We have revised the English editing, according to the reviewer suggestion.
7. Pag. 8 – Line 296: lessening.
7. We have changed the words “secondly to” with the word “lessening” as suggested by reviewer.
8. Pag. 8 – Line 302: in order to.
8. We have changed the word “to” with the words “in order to” as suggested by reviewer.
Reviewer 2 Report
Could you clarify the use of the ICSD-II, instead of the ICSD-III?
Your age cut-off for the definition of elderly is 60. Would you r teview have been different if you chose 70? 55?
Can you comment on how these methods may differ fro those that have targeted a younger or a more general audience>
The section on "Sleep Technologies" seems irrelevant to the other points in the paper.
Author Response
1. Could you clarify the use of the ICSD-II, instead of the ICSD-III?
1. We have considered the ICSD-II in order to extend the literature research.
2. Your age cut-off for the definition of elderly is 60. Would you review have been different if you chose 70? 55?
2. The age cut-off was ≥ 60 years (so comprising elders with 70 years too). Surely, if we have comprising people with ≥ 55 years, we have achieved only three studies plus.
3. Can you comment on how these methods may differ fro those that have targeted a younger or a more general audience
3. We are geriatricians, and we have interest to address at a older user group.
4. The section on "Sleep Technologies" seems irrelevant to the other points in the paper.
4. We have decided to let the aforesaid paragraph because it is not assimilable to other paragraphs.